

# Simulating the dust emissions and SOA formation over Northern Africa during the mid-Holocene Green Sahara period

Putian Zhou[1,*], Zhengyao Lu[2], Jukka-Pekka Keskinen[3], Qiong Zhang[4], Juha Lento[5], Jianpu Bian[1], Twan van Noije[6], Philippe Le Sager[6], Veli-Matti Kerminen[1], Markku Kulmala[1], Michael Boy[1,7], Risto Makkonen[1,3,*]

[1]Institute for Atmospheric and Earth System Research/Physics, Faculty of Science, University of Helsinki, Finland
[2]Department of Physical Geography and Ecosystem Science, Lund University, Sweden
[3]Finnish Meteorological Institute, FI-00560, Finland
[4]Department of Physical Geography and Bolin Centre for Climate Research, Stockholm University, Sweden
[5]CSC - IT Center for Science, Finland
[6]Royal Netherlands Meteorological Institute (KNMI), Netherlands
[7]School of Engineering Sciences, Lappeenranta-Lahti University of Technology LUT, Finland

*Correspondence to*: Putian Zhou (putian.zhou@helsinki.fi) and Risto Makkonen (risto.makkonen@fmi.fi)

**Abstract.** Paleo-proxy data indicates that a "Green Sahara" thrived in Northern Africa during the early- to mid-Holocene (MH; 11,000 to 5,000 years before present), characterized by more vegetation cover and reduced dust emission. Utilizing a state-of-the-art atmospheric chemical transport model TM5-MP, we assessed the changes in biogenic volatile organic compounds (BVOCs) emissions, dust emission and secondary organic aerosol (SOA) concentration in Northern Africa during this period relative to the pre-industrial (PI) period. Our simulations show that dust emissions reduced from 280.6 Tg a$^{-1}$ in the PI to 26.8 Tg a$^{-1}$ in the MH, agreeing with indications from eight marine sediment records in the Atlantic Ocean. The northward expansion in Northern Africa resulted in an increase in annual emissions of isoprene and monoterpenes during the MH, around 4.3 and 3.5 times higher than that in the PI period, respectively, causing 1.9 times increase in the SOA surface concentration. The enhanced SOA surface concentration and decreased sulfate surface concentration counteracted each other, leading to a 17% increase in the cloud condensation nuclei at 0.2% super saturation over Northern Africa. Our simulations provide consistent emission datasets of BVOCs, dust, and the SOA formation aligned with the northward shift of vegetation during the "Green Sahara" period, which could serve as a benchmark for MH aerosol input in future Earth system model simulation experiments.

## 1 Introduction

A Green Sahara, instead of present desert Sahara, existed in Northern Africa during the early- to mid-Holocene (MH) from 11 to 5 kaBP (thousand years before present which is 1950, ka) which is usually referred to African Humid Period (AHP, Claussen et al., 2017). Fossil pollen records suggest that tropical plants expanded northward up to 23° N (Hély et al., 2014), indicating a shift in vegetation patterns. Leaf wax analysis reveals a striking increase in annual precipitation in the western



Sahara, from the present-day range of 35 to 100 mm to a substantial 640 mm during the AHP (Tierney et al., 2017). Paleohydrological records also depict a landscape dotted with lakes across Northern Africa, extending at least to 28° N as

shown by Lézine et al. (2011). These findings collectively paint a vivid picture of a once green and flourishing Sahara.

The higher summer insolation in the Northern Hemisphere due to Earth's orbital variation is considered to intensify the West African Monsoon (WAM), leading to increased rainfall across Northern Africa (Kutzbach, 1981). However, this theory is insufficient to explain the substantial rise in rainfall needed to maintain the vegetation cover over Northern Africa, as shown

in model simulations (Joussaume et al., 1999). The northward vegetation shift is correlated with a reduction of air-borne mineral dust, suggesting that land cover change could contribute to the stronger WAM and enhanced rainfall. Pausata et al. (2016) demonstrated that considering reduced atmospheric dust levels could potentially push the WAM an additional 500 km northward under identical vegetation change. Egerer et al. (2018) showed improved rainfall modelling over Northern Africa when introducing dynamic vegetation and interactive dust, compared to CMIP5 model results. Nevertheless, including the

indirect effects of dust aerosol on stratiform rainfall could reduce dust-induced Saharan precipitation anomaly by about 13% (Thompson et al., 2019). Other hypotheses were also proposed. For example, Menviel et al. (2021) suggested that the abrupt onset of AHP was mainly due to the strengthening of Atlantic Meridional Overturning Circulation (AMOC).

Vegetation, beyond altering the surface albedo and the dust emissions, emits large quantities of biogenic volatile organic

compounds (BVOCs) (Guenther et al., 2006; Guenther et al., 2012). These BVOCs play a key role in biosphere-atmosphere interactions, they can be oxidized to low-volatile compounds to form secondary organic aerosol (SOA). SOA then modulate radiative forcing directly by scattering or absorbing shortwave radiation, and indirectly by acting as cloud condensation nuclei (CCN) (Shrivastava et al., 2017). In the context of a Green Sahara, reduced dust emissions could enhance SOA formation efficiently due to lower condensation sinks, and simultaneously reduce the giant CCN concentration impeding the

initiation of precipitation (Carslaw et al., 2010). Therefore, developing a comprehensive dataset of BVOC emissions, dust emissions and SOA concentration that is consistent with the reconstructed vegetation cover, is important to understand Northern Africa's climate change during the MH.

To date, this area has been relatively unexplored. Adams et al. (2001) reconstructed vegetation biome distribution and

estimated the global BVOC emissions based on reconstructed vegetation biome distribution from 8 ka and 5 ka. Kaplan et al. (2006) examined BVOC impacts on MH methane concentration using a global vegetation model, but failed to reproduce the northward vegetation shift over the Sahara. Therefore, in this study we aim to use state-of-the-art atmospheric chemical transport model simulations to provide the most comprehensive datasets of BVOC emissions, dust emissions and SOA concentration in Northern Africa, based on the consistently accurate vegetation simulation result from Lu et al. (2018). We

also examined the consequential climate-relevant changes in aerosol optical properties and CCN to reveal their driving factors.



This study is organized as follows. In Section 2, the model TM5-MP is introduced and the detailed model configurations are described. This section also includes the description of the datasets of BVOC emissions from various sources, as well as how
the reconstructed dust mass deposition flux data at different marine sediment locations are obtained. In Section 3, several aerosol-related quantities over Africa, especially Northern Africa, during PI and MH periods are analyzed, e.g., the BVOC emissions, dust emissions and deposition fluxes, the surface concentrations of SOA and CCN, and aerosol optical depth at 550 nm. The results are concluded in Section 4.

## 2 Methods and data

### 2.1 TM5-MP

The global chemical transport model TM5-MP (Tracer Model 5, Massively Parallel version; Krol et al., 2005; Huijnen et al., 2010; van Noije et al., 2014; Williams et al., 2017; van Noije et al., 2021) has been applied in this study. It is driven by the input meteorological fields which are produced from ERA-Interim reanalysis datasets provided by the ECMWF (European Centre for Medium-range Weather Forecasts; Dee et al., 2011). The model accounts for gas-phase, aqueous-phase, and
heterogeneous chemistry. The gas-phase chemistry scheme is a modified version of the CB05 carbon bond mechanism (Yarwood et al., 2005), which was described in details in Williams et al. (2017).

TM5-MP predicts aerosol dynamic processes using the two-moment modal model M7 (Vignati et al., 2004), which includes seven log-normally distributed modes with four water-soluble modes (nucleation, Aitken, accumulation and coarse) and
three insoluble modes (Aitken, accumulation and coarse). Each mode corresponds to a specific dry diameter range: nucleation mode is less than 10 nm, Aitken mode ranges from 10 nm to 100 nm, accumulation mode from 100 nm to 1 μm, and coarse mode is greater than 1 μm.

The current version of TM5-MP incorporates SOA into four soluble modes and the insoluble Aitken mode, resulting in a
comprehensive aerosol model including SOA, sulfate, ammonium, nitrate, methane sulfonic acid (MSA), primary organic aerosol, black carbon, sea salt, and mineral dust (Bergman et al., 2022). Isoprene and monoterpenes, acting as gaseous precursors of SOA, are from natural emissions calculated by MEGANv2.1 (Model of Emissions of Gases and Aerosols from Nature version 2.1; Guenther et al., 2012; Sindelarova et al., 2014) and biomass burning emissions based on van Marle et al. (2017). The isoprene and monoterpenes can be oxidized by OH and $O_3$ in the air to formextremely low volatility organic
compounds (ELVOCs) and semi-volatile organic compounds (SVOCs). The ELVOCs will first participate in new particle formation and the rest will condense onto existing aerosol particles. The SVOCs will only condense due to their higher volatilities. The particle formation and condensation processes occur within one time step, so the transport of ELVOCs and SVOCs is not simulated. More details were described in Bergman et al. (2022).



The dust emissions are calculated online based on the method which was introduced in Tegen et al. (2002) and extended by
Heinhold et al. (2007). New dust emission parameterization methods have also been developed (e.g., Leung et al., 2023) but
have not been implemented into current TM5-MP version. Only the features related to this study are described here. Firstly,
we proportion dust emission to the non-vegetated area within a grid cell, which is determined by the vegetation cover data
(see Eq. 1 in Tegen et al., 2002). When dominated by shrubs, we apply the maximum annual vegetation cover, whereas when

grass dominates, we use the monthly value. Secondly, Tegen et al. (2002) simulated the potential maximum extent of lakes,
and determined preferential dust source regions from the difference between the maximum areas and the actual present-day
lakes. These regions are assumed to contain silt aggregates composed of finer particles with median particle radius of 15 μm
in the Northern Hemisphere (NH) and 27 μm in the Southern Hemisphere (SH), requiring a smaller wind stress threshold to
lift, which is 30 cm s$^{-1}$ and 20 cm s$^{-1}$ in the NH and SH, respectively. Moreover, the ratio between the vertical dust flux and

horizontal soil particle flux is also assumed to be the largest among all the regions, which is set to 10$^{-5}$ cm$^{-1}$. All these factors
have determined high dust emission fluxes over the preferential source regions. However, we should note that in TM5-MP,
the threshold wind stress ($u_{min}$) is set to 13.75 cm s$^{-1}$ for all kinds of land covers, which reduce further when the cultivation
cover exceeds 8% and the grid cell is dominated by either grass or shrub. For more details refer to Tegen et al. (2002) and
van Noije et al. (2021).

**2.2 Model set-up**

Simulations have been conducted for five different cases: a PI control run (pi_ctrl), a PI run with the default TM5-MP
configuration (pi_orig), a PI control run excluding anthropogenic emissions (pi_zero), a standard MH run (mh) and a MH
run incorporating a Green Sahara and reduced dust (mh_gsrd) (Table 1). The model configuration for each case will be
detailed subsequently. All simulations applied a horizontal resolution of 3 degrees in longitude and 2 degrees in latitude with

34 vertical hybrid-sigma levels. The base time step was set to one hour. Each simulation ran for two years, with the first year
serving as the spin-up and the results from the second year were analyzed. The model was installed on Puhti supercomputer
at CSC (IT Center for Science, Finland), and 90 CPU (Central Processing Unit) cores were utilized for each parallel
simulation run. One simulation year cost about 10 hours in real life.

Unless otherwise specified, the input meteorological data for spin-up simulations were derived from the year 2008 and the
year 2009 was used for subsequent-year simulations, facilitating a focus on comparisons between individual cases. The
vegetation cover data for pi_ctrl, mh and mh_gsrd cases were derived from respective PI, MH and MH_gsrd simulation
results in Lu et al. (2018).



*Table 1: Model configurations of the simulation cases pi_ctrl, pi_orig, pi_zero, mh and mh_gsrd. The number 1850 and 2009 show the years that the input data are from. Here lsm means land-sea mask, PD is present day lsm and MH is mid-Holocene lsm with paleolakes switched on as water surface. For vegetation, pi_mean, mh_mean and mh_gsrd_mean represent the 10-year mean of the simulation data from PI, MH and MH_gsrd cases in Lu et al. (2018), respectively. For the emissions of isoprene and monoterpes, as well as the FPAR (absorbed photosynthetically active radiation), pi_mean, mh_mean and mh_gsrd_mean represent that the data are derived from their corresponding vegetation data. The "original" means using default original TM5 input data. The mixing ratios of $CO_2$ and $CH_4$ are fixed to 265.4 ppmv and 597.0 ppbv, respectively for MH cases.*

|  |  | pi_ctrl | pi_orig | pi_zero | mh | mh_gsrd |
|---|---|---|---|---|---|---|
| **METEO** | **general** | 2009 | 2009 | 2009 | 2009 | 2009 |
|  | **lsm** | PD | PD | PD | MH | MH |
|  | **veg** | pi_mean | 2009 | pi_mean | mh_mean | mh_gsrd_mean |
| **EMISS** | **isoprene** | pi_mean | 2009 | pi_mean | mh_mean | mh_gsrd_mean |
|  | **monoterpenes** | pi_mean | 2009 | pi_mean | mh_mean | mh_gsrd_mean |
|  | **other natural** | 2009 | 2009 | 2009 | 2009 | 2009 |
|  | **anthropogenic** | 1850 | 1850 | zero | zero | zero |
| **onlinedust** | **paleolakes** | original | original | original | on | on |
|  | **FPAR** | pi_mean | original | pi_mean | mh_mean | mh_gsrd_mean |
|  | **cultivation** | zero | original | zero | zero | zero |
| **CO2** |  | 1850 | 1850 | 1850 | 264.4 ppmv | 264.4 ppmv |
| **CH4** |  | 1850 | 1850 | 1850 | 597.0 ppbv | 597.0 ppbv |

The anthropogenic emissions are set to 1850 levels for pi_orig and pi_ctrl cases, and are switched off for mh and mh_gsrd
cases. The natural emissions in 2009 are applied in pi_ctrl, pi_orig, mh and mh_gsrd cases, except that the emissions of isoprene and monoterpenes in pi_ctrl, mh and mh_gsrd cases are derived from the LPJ-GUESS (Lund-Potsdam-Jena General Ecosystem Simulator; Smith et al., 2001; Smith et al., 2014) simulation results of the cases PI, MH and MH_gsrd in Lu et al. (2018), respectively. Given the simulation results from Lu et al. (2018) contained 10-year of equilibrium-state data, we applied 10-year mean for both the vegetation and BVOC emission data (see supplement). In mh and mh_gsrd cases, the
mixing ratios of greenhouse gases $CH_4$ and $CO_2$ were set to 597.0 ppbv and 264.4 ppmv respectively according to the PMIP4-CMIP6 protocol for MH experiments (Otto-Bliesner et al., 2017). The configuration of natural emissions of isoprene and monoterpenes as well as the vegetation cover data in pi_ctrl, mh and mh_gsrd cases was applied globally.





The parameters used to calculate the dust emissions in pi_ctrl, mh and mh_gsrd cases are modified according to their input vegetation data and specific time period assumptions. Firstly, all the preferential dust emission source areas, namely potential lake areas, within the region of northern Africa (20° W to 40° E and 10° N to 30° N) are switched off in the mh and mh_gsrd cases, since we assume that these areas were actual lakes during MH period. A similar assumption was also applied in Egerer et al. (2018). Secondly, the land-sea mask ratio over these areas has also been set to zero, representing lake surface within these grids. Thirdly, the cultivation cover was set to zero for the mh and mh_gsrd cases, and for comparison, we made the same settings for the pi_ctrl case. Fourthly, within the TM5-MP model, vegetation cover impacting the dust emission fluxes is indirectly represented by FPAR (the fraction of absorbed photosynthetically active radiation) according to Tegen et al. (2002). Here we applied the monthly vegetation cover directly from LPJ-GUESS output data within the Northern Africa region, and set the corresponding values to the FPAR variable for pi_ctrl, mh and mh_gsrd cases. The pi_orig applied the original model set-up for the PI but with present-day meteorological input data. The configuration of case pi_zero is the same as pi_ctrl except the anthropogenic emissions are switched off.

### 2.3 BVOC emission data from other sources

In order to compare our results with the simulated BVOC emissions from other models, we downloaded the model data from the *midHolocene* experiment in PMIP4-CMIP6 (CMIP, 2023), including *emivoc* (total emission rates of isoprene and monoterpenes) from NorESM2-LM, and *emiisop* (emission rate of isoprene) from both NorESM2-LM and MRI-ESM2-0. Currently, these are the only available BVOC emission data during MH published in PMIP4-CMIP6 database. The emission rate of monoterpenes from NorESM2-LM was calculated by substracting *emiisop* from *emivoc*.

Furthermore, other available data from previous studies were also collected here (Table 2). Adams et al. (2001) estimated the global total annual emission rates of isoprene and monoterpenes in 5 ka which were 666.5 Tg a$^{-1}$ and 137.6 Tg a$^{-1}$, respectively. Under the scenario of "present potential vegetation" (representing the PI vegetation), they estimated the isoprene emission rate as 561.4 Tg a$^{-1}$ and monoterpenes emission rate as 116.5 Tg a$^{-1}$. Kaplan et al. (2006) simulated the emission rates of isoprene and monoterpenes as 516.4 TgC a$^{-1}$ (585.3 Tg a$^{-1}$) and 117.6 TgC a$^{-1}$ (133.3 Tg a-1) for 6 ka, 540.7 TgC a$^{-1}$ (612.8 Tg a$^{-1}$) and 121.3 TgC a$^{-1}$ (137.5 Tg a$^{-1}$) for PI, respectively. Singarayer et al. (2011) calculated the time series of global isoprene emission rate over the last glacial cycle, and we estimated the value in 6 ka as about 860 TgC a$^{-1}$ (974.7 Tg a$^{-1}$) and in present-day as about 950 TgC a$^{-1}$ (1076.7 Tg a$^{-1}$) from their Fig. 1d.



*Table 2: Global or regional averaged or summed values of different quantities for the simulation cases pi_ctrl, pi_zero (only for sconcccn020), mh and mh_gsrd, as well as other data sources, including noresm (NorESM2-LM PMIP4 midHolocene experiment), mriesm (MRI-ESM2-0 PMIP4 midHolocene experiment), A2001 (Adams et al., 2001), K2006 (Kaplan et al., 2006) and S2011 (Singarayer et al., 2011). The values listed are: area summed annual values of emiisop (isoprene emission rate, Tg a$^{-1}$), emiterp (emission rate of monoterpenes, Tg a$^{-1}$), emidust (dust emission rate, Tg a$^{-1}$), depdust (dust deposition rate, Tg a$^{-1}$); area summed annual values of loaddust (dust load, Tg), loadsoa (SOA load, Tg); area averaged annual mean values of sconcdust (surface mass concentration of dust, µg m$^{-3}$), sconcsoa (surface mass concentration of SOA, µg m$^{-3}$), sconcccn020 (surface concentration of CCN at supersaturation of 0.20%, # cm$^{-3}$), od550aer (aerosol AOD at 550 nm), od550soa (SOA AOD at 550 nm), and od550dust (dust AOD at 550 nm).*

| Variable name | Case | Region | | | |
|---|---|---|---|---|---|
| | | Global | Northern Africa | Lake Chad | Western Sahara catchments |
| emiisop | pi_ctrl | 688.0 | 27.0 | 3.4 | 1.6 |
| | mh | 757.6 | 50.5 | 9.2 | 3.7 |
| | mh_gsrd | 889.7 | 114.8 | 13.5 | 16.5 |
| | noresm | 446.4 | 0.81 | 0.090 | 0.0043 |
| | mriesm | 466.3 | 7.3 | 1.5 | 0.073 |
| | A2001, PI | 561.4 | NA | NA | NA |
| | A2001, 5 kaBP | 666.5 | NA | NA | NA |
| | K2006, PI | 612.8 | NA | NA | NA |
| | K2006, 6 kaBP | 585.3 | NA | NA | NA |
| | S2011, PD | 1076.7 | NA | NA | NA |
| | S2011, 6 kaBP | 974.7 | NA | NA | NA |
| emiterp | pi_ctrl | 61.4 | 2.3 | 0.31 | 0.16 |
| | mh | 67.5 | 3.9 | 0.70 | 0.37 |
| | mh_gsrd | 74.3 | 8.0 | 0.79 | 1.5 |
| | noresm | 96.8 | 0.12 | 0.0060 | 0.00026 |
| | mriesm | NA | NA | NA | NA |
| | A2001, PI | 116.5 | NA | NA | NA |
| | A2001, 5 kaBP | 137.6 | NA | NA | NA |
| | K2006, PI | 137.5 | NA | NA | NA |
| | K2006, 6 kaBP | 133.3 | NA | NA | NA |
| depdust | pi_ctrl | 855.4 | 135.0 | 36.4 | 28.4 |
| | mh | 664.8 | 54.0 | 5.1 | 12.7 |





| | | | | | |
|---|---|---|---|---|---|
| | **mh_gsrd** | 594.8 | 21.3 | 2.5 | 4.0 |
| **emidust** | **pi_ctrl** | 854.9 | 280.6 | 80.1 | 82.1 |
| | **mh** | 665.9 | 95.0 | 2.8 | 25.2 |
| | **mh_gsrd** | 596.1 | 26.8 | 2.9 | 5.2 |
| **sconcdust** | **pi_ctrl** | 4.7 | 89.0 | 203.2 | 99.6 |
| | **mh** | 3.2 | 33.1 | 31.8 | 38.0 |
| | **mh_gsrd** | 2.8 | 15.9 | 15.6 | 15.1 |
| **loaddust** | **pi_ctrl** | 9.2 | 2.1 | 0.41 | 0.45 |
| | **mh** | 6.1 | 0.77 | 0.10 | 0.17 |
| | **mh_gsrd** | 5.2 | 0.37 | 0.044 | 0.068 |
| **sconcsoa** | **pi_ctrl** | 3.2 | 1.0 | 1.3 | 0.80 |
| | **mh** | 3.6 | 1.5 | 2.2 | 1.2 |
| | **mh_gsrd** | 4.3 | 2.9 | 3.9 | 2.8 |
| **loadsoa** | **pi_ctrl** | 1.3 | 0.066 | 0.0096 | 0.011 |
| | **mh** | 1.4 | 0.091 | 0.014 | 0.015 |
| | **mh_gsrd** | 1.6 | 0.15 | 0.023 | 0.029 |
| **sconcccn020** | **pi_zero** | 0.00051 | 96.2 | 110.2 | 74.1 |
| | **pi_ctrl** | 0.00063 | 127.7 | 138.7 | 99.4 |
| | **mh** | 0.00053 | 103.2 | 120.6 | 79.6 |
| | **mh_gsrd** | 0.00054 | 112.6 | 130.3 | 92.0 |
| **od550aer** | **pi_ctrl** | 6.8e-7 | 0.212 | 0.323 | 0.211 |
| | **mh** | 6.2e-7 | 0.122 | 0.146 | 0.116 |
| | **mh_gsrd** | 6.4e-7 | 0.132 | 0.156 | 0.124 |
| **od550dust** | **pi_ctrl** | 1.6e-7 | 0.147 | 0.247 | 0.155 |
| | **mh** | 1.1e-7 | 0.048 | 0.054 | 0.051 |
| | **mh_gsrd** | 9.6e-8 | 0.026 | 0.027 | 0.024 |
| **od550soa** | **pi_ctrl** | 1.2e-7 | 0.032 | 0.040 | 0.027 |
| | **mh** | 1.3e-7 | 0.045 | 0.059 | 0.038 |
| | **mh_gsrd** | 1.6e-7 | 0.077 | 0.096 | 0.074 |

As a comparison, Arneth et al. (2008) summarized previous studies and showed that the estimated ranges of present-day global total annual emissions of isoprene and monoterpenes are from 412 TgC a$^{-1}$ (467 Tg a$^{-1}$) to 601 TgC a$^{-1}$ (681 Tg a$^{-1}$), and from 32 TgC a$^{-1}$ (36 Tg a$^{-1}$) to 127 TgC a$^{-1}$ (144 Tg a$^{-1}$), respectively. More recent studies, including the estimation of present-day global total annual BVOC emissions, present a varied picture. Hantson et al. (2017), for example, showed an



estimate of 385 TgC a$^{-1}$ (436 Tg a$^{-1}$) and 28.6 TgC a$^{-1}$ (32.4 Tg a$^{-1}$) for isoprene and monoterpenes. Meanwhile, Cao et al. (2021) presented a range of 411 to 473 TgC a$^{-1}$ (466 to 536 Tg a$^{-1}$) for isoprene emissions across all CMIP6 models. Sindelarova et al. (2022) modelled various BVOC species in different CAMS-GLOB-BIO inventory versions, showing an isoprene emission range of 299.1 Tg a$^{-1}$ to 440.5 Tg a$^{-1}$ and a monoterpenes emission range of 63.2 Tg a$^{-1}$ to 82.7 Tg a$^{-1}$. The molar masses of carbon, isoprene, monoterpenes are assigned to 12 g mol$^{-1}$, 68 g mol$^{-1}$ and 136 g mol$^{-1}$ in the unit conversion between TgC and Tg.

## 2.4 Marine sediment record

The reconstructed dust mass deposition flux data at different marine sediment sites in North Atlantic were collected for the comparison with model results. The sediment sites are shown in Fig. 1 and Table 3 with GC37, GC49, GC68 and ODP658C along the northwest African coast, VM20-234, GGC3 and GGC6 in the remote Atlantic Ocean, and 103GGC in the Bahamas. The $^{230}$Th (Thorium) normalization method was applied to calculate the dust fluxes at GC37, GC49, GC68 (McGee et al., 2013; Albani et al., 2015), ODP658C (Adkins et al., 2016), 103GGC and VM20-234 (Williams et al., 2016). The dust flux values at ODP658C were not given directly in the original references, so the terrigenous flux ($F_{terr}$) is calculated here with the original excess $^{230}$Th (Ex$^{230}$Th) data according to the method in Adkins et al. (2016):

$$F_{terr} = \frac{\beta z}{Ex^{230}Th} \cdot f_{terr} , \tag{1}$$

where $\beta$ is the production rate of $^{230}$Th from $^{234}$U in the water column, equal to $2.63 \times 10^{-5}$ dpm cm$^{-3}$ ka$^{-1}$, and dpm represents disintegration per minute which is a radioactive decay unit. Here z is the water depth of the sediment core location, which is 2263 m (deMenocal et al., 2000), $f_{terr}$ is the fraction of the terrigenous flux out of the total sediment flux, which is set to 45% for MH period and 60% for PI period (see Table 2 in deMenocal et al., 2000). It should be noted that the terrigenous flux, which includes both the fluvial/shelf-derived and the eolian terrigenous flux, typically exceeds the dust flux. Therefore, we only compared the latter one with the model results (McGee et al., 2013). Similarly, the dust flux values at GGC3 and GGC6 were also not provided directly in Middleton et al. (2018), so these values ($F_{dust}$) were calculated from the original data as:

$$F_{dust} = \frac{[^{4}He_{terr}] \cdot vsr}{5600} , \tag{2}$$

where [$^{4}He_{terr}$] is the amount of terrigenous $^{4}$He (Helium) in the sediment in the unit of ncc g$^{-1}$, and ncc is equal to 10$^{-9}$ cm$^{3}$ at standard temperature and pressure. And vsr is vertical sediment rain rate in the unit of g cm$^{-2}$ ka$^{-1}$, 5600 ncc g$^{-1}$ is applied to convert the $^{4}$He to $^{4}$He-based dust fluxes.





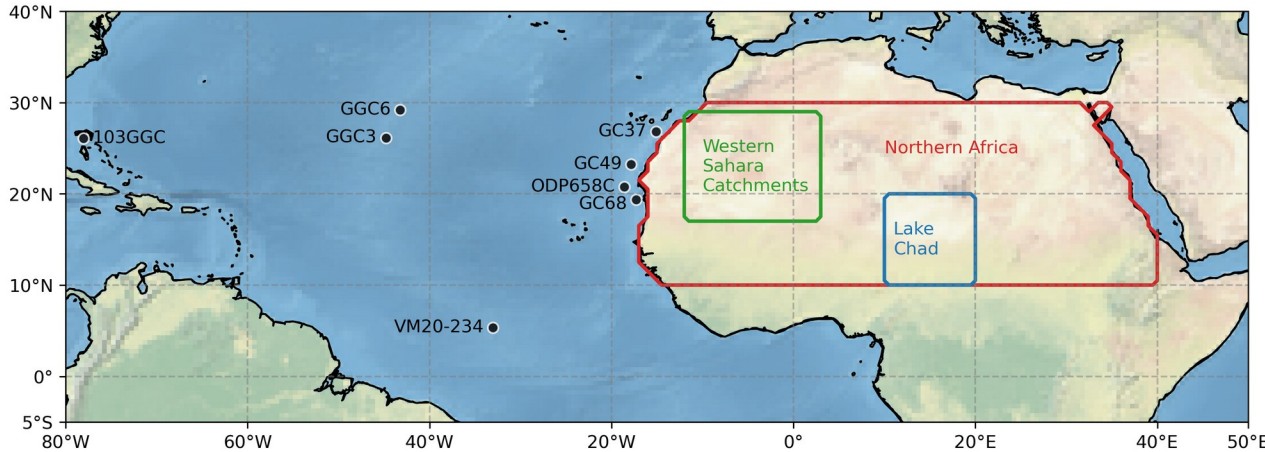

*Figure 1: The marine sediment sites and the regions. The exact location of the sites are listed in Table 3. The red line encloses the northern Arica (NA) region, which is defined as African continent area within the region 20° W to 40° E and 10° N to 30° N. The blue box shows the Lake Chad (LC) region within 10° E–20° E and 10° N–20° N, the green box shows western Sahara catchments (WSC) region around the paleo Lake Timbuktu area within 12° W–3° E and 17° N–29° N.*

Since the reconstructed sample ages are not exactly at PI (0.1 ka, 1850 compared to 1950) or MH (6 ka assumed in our model simulations), the data closest to 0.1 ka and 6 ka were selected. For the PI period, only the data younger than 1 ka were selected. For example, for GC49, the data in 0.55 ka was selected for PI and the data in 5.57 ka and 6.57 were selected for the MH period. Moreover, the estimated errors of one standard deviation of age and dust flux data were also collected whenever they were available in the literatures.

**2.5 Study domains**

In this study, we defined three regions for detailed analysis besides the globe scale. Firstly, the Northern Africa (NA) is defined as the African continent area within the region 20° W to 40° E and 10° N to 30° N. The other two regions are related to the paleo water bodies during MH where the dust emissions were reduced the most. One is the Lake Chad (LC) area within 10° E - 20° E and 10° N - 20° N, the other is Western Sahara Catchments (WSC) centered around the paleo Lake Timbuktu area within 12° W - 3° E and 17° N - 29° N. The two paleo lakes are selected because the Lake Chad was suggested to thrive between 11 ka and 5 ka (Armitage et al., 2015), and Lake Timbuktu was considered to experience a wet period between 9.5 ka and 3.5 ka (Drake et al., 2022). The regions on the map are shown in Fig. 1. And the regional averaged and summed values in Table 2 are also referring to the regions defined here.





*Table 3: Reconstructed and modelled dust mass deposition fluxes (DMDF) at different sites. The references of reconstructed data are listed below the Table. The estimated errors of reconstructed data are shown in the brackets if they are provided in the original references.*

| Site | Lat (° N) | Lon (° W) | Reconstructed data age (ka BP) | Reconstructed DMDF (g m⁻² yr⁻¹) | Modelled DMDF (g m⁻² yr⁻¹) | | |
|---|---|---|---|---|---|---|---|
| | | | | | pi_ctrl | mh | mh_gsrd |
| GC37[1] | 26.82 | 15.12 | 6.22 (±0.23) | 0.92 (±0.14) | 1.56 | 0.94 | 0.32 |
| GC49[1] | 23.21 | 17.85 | 0.55 (±0.24) | 5.70 (±0.89) | 5.51 | 4.21 | 0.78 |
| | | | 5.57 (±0.34) | 1.11 (±0.17) | | | |
| | | | 6.57 (±0.08) | 1.23 (±0.19) | | | |
| GC68[1] | 19.36 | 17.28 | 5.89 (±0.14) | 4.00 (±0.68) | 16.56 | 11.36 | 3.70 |
| | | | 6.30 (±0.24) | 3.67 (±0.65) | | | |
| ODP658C[2] | 20.75 | 18.58 | 0.193 | 17.990 | 10.63 | 7.66 | 2.20 |
| | | | 5.808 | 8.554 | | | |
| | | | 6.057 | 8.901 | | | |
| VM20-234[3] | 5.33 | 33.03 | 0.97 | 2.96 (±0.13) | 4.40 | 1.22 | 0.66 |
| | | | 6.03 | 1.66 (±0.07) | | | |
| 103GGC[3] | 26.07 | 78.06 | 0.29 | 0.82 (±0.06) | 0.86 | 0.43 | 0.22 |
| | | | 5.88 | 0.53 (±0.04) | | | |
| GGC3[4] | 26.14 | 44.80 | 6.0 | 0.22 | 0.86 | 0.34 | 0.13 |
| GGC6[4] | 29.21 | 43.23 | 0.5 | 0.29 | 0.57 | 0.26 | 0.15 |
| | | | 6.2 | 0.14 | | | |

[1]McGee et al., 2013, Albani et al., 2015.

[2]deMenocal et al., 2000, Adkins et al., 2006

[3]Williams et al., 2016

215    [4]Middleton et al., 2018

## 3 Results and discussions

### 3.1 Surge in BVOC emissions

In the absence of available paleo proxies to directly derive the BVOC emission data, we must rely on model simulations. Currently, the only available model results of BVOC emissions during MH from the PMIP4-CMIP6 midHolocene

220    experiments come from NorESM2-LM  and MRI-ESM2-0 simulations as mentioned above. Figure 2 shows the total annual



emission rates of isoprene and monoterpenes from simulation cases pi_ctrl, mh and mh_gsrd in present study, along with the results from NorESM2-LM and MRI-ESM2-0.

Due to the northward extension of vegetation cover over the region NA in the cases mh and mh_gsrd (elucidated in Lu et al., 2018), both the emissions of isoprene and monoterpenes increase compared to the pi_ctrl case (Fig. 2a, b, c, f, g, h). Notably, the BVOC emissions nearly cover the entire NA region in mh_gsrd (Fig. 2c, h). The total isoprene emission over the region NA is 27.0 Tg a$^{-1}$, 50.5 Tg a$^{-1}$ and 114.8 Tg a$^{-1}$ in pi_ctrl, mh and mh_gsrd, respectively. Correspondingly, the monoterpenes emission rate is 2.3 Tg a$^{-1}$, 3.9 Tg a$^{-1}$ and 8.0 Tg a$^{-1}$, respectively (Table 2). The relative increase in mh compared to pi_ctrl is 87% for isoprene emission and 70% for monoteprenes emission, respectively. The mh_gsrd case, on the other hand, marks a more dramatic surge, with isoprene and monoterpenes emissions increasing to 4.3 times and 3.5 times of pi_ctrl, respectively.

Compared with the mh and mh_gsrd cases, the NorESM2-LM and MRI-ESM2-0 show lower isoprene emission rates over NA than the pi_ctrl case (Fig. 2a, d, e), which are 0.81 Tg a$^{-1}$ and 7.3 Tg a$^{-1}$, respectively. Moreover, the BVOC emissions in these two models are mostly centered over central Africa, encompassing the present-day Congolian rainforest (Fig. 2d, e, i). In contrast, in our cases pi_ctrl, mh and mh_gsrd in which the vegetation cover was simulated by the vegetation dynamic model LPJ-GUESS, the BVOC emissions are more evenly distributed over Central and Southern Africa, which remain substantial at the southern end of Africa, reaching 0.4 μg m$^{-2}$ s$^{-1}$ for isoprene emission and 0.03 μg m$^{-2}$ s$^{-1}$ for monoterpenes emission, respectively.

Besides the PMIP4-CMIP6 midHolocene experiment results, several previous studies also provided estimates of BVOC emissions during the MH as mentioned in Sec. 2.3. Among them, only Adams et al. (2001) reported a rise in isoprene emission during MH compared to PI or present-day levels, in which the global total annual emission rate of isoprene increased by 19%, and for monoterpenes by 18% given the reconstructed vegetation cover. Moreover, in Fig. 1a and 1d in Adams et al. (2001) we can see that the area with isoprene emission rate larger than 2 Tg Mkm$^{-2}$ a$^{-1}$ (about 0.06 μg m$^{-2}$ s$^{-1}$, here the unit in Fig. 1 in Adams et al. (2001) is Tg Mha$^{-1}$ a$^{-1}$, but it should be a typo and the correct unit is Tg Mkm$^{-2}$ a$^{-1}$) moves northward over NA in 5 ka, and the area with larger than 15 Tg Mkm$^{-2}$ a$^{-1}$ (about 0.48 μg m$^{-2}$ s$^{-1}$) increased in the Sahel region.





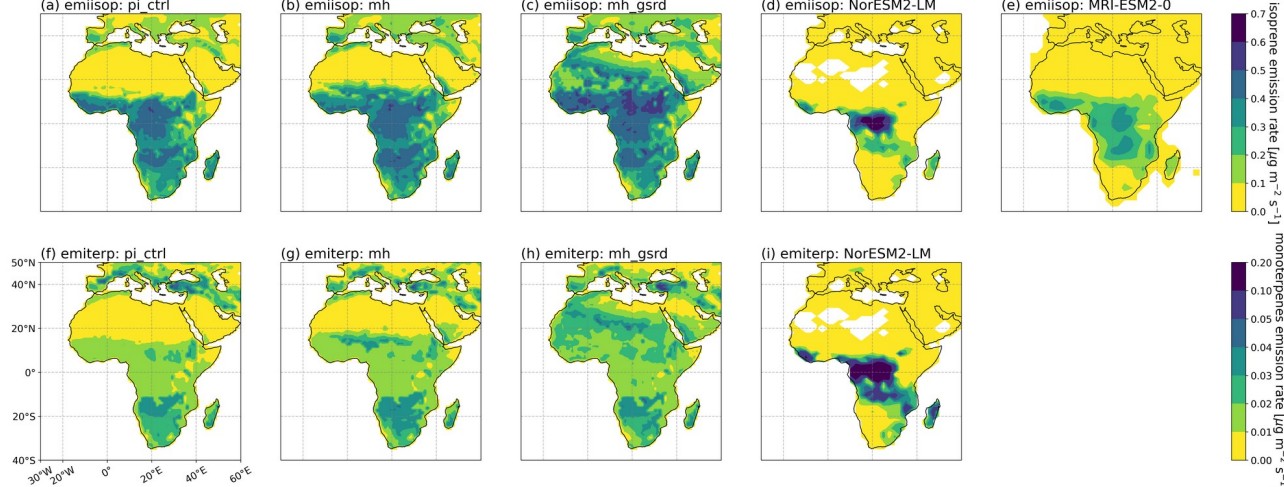

*Figure 2: Annual mean emission rates of isoprene (emiisop) over Africa from the simulation cases pi_ctrl (a), mh (b) and mh_gsrd (c), as well as the PMIP4 simulation results of NorESM2-LM (d) and MRI-ESM2-0 (e). The annual mean emission rates of monoterpenes (emiterp) are also shown for simulation cases pi_ctrl (f), mh (g), mh_gsrd (h) and the simulation results of NorESM2-LM (i).*

In this study, the total annual global emissions of isoprene and monoterpenes are 889.7 Tg a$^{-1}$ and 74.3 Tg a$^{-1}$ in the mh_gsrd case, 688.0 Tg a$^{-1}$ and 61.4 Tg a$^{-1}$ in the pi_ctrl case, respectively. These values align with the higher BVOC emissions during the MH similar with that in Adams et al. (2001). On the contrary, Kaplan et al. (2006) noted lower BVOC emissions in 6 ka, with 585.3 Tg a$^{-1}$ for isoprene and 133.3 Tg a$^{-1}$ for monoterpenes compared to 612.8 Tg a$^{-1}$ and 137.5 Tg a$^{-1}$ during the PI period. They also specified that their vegetation simulations could not reproduce the northward shift of vegetation
over Sahara region, likely due to their climate model (HC-UM, Hadley Centre Unified Model) inable to capture the interactive processes between land, atmosphere and ocean. Singarayer et al. (2011) also showed a decreasing trend of global isoprene emission rate from the PI period to the MH period. Moreover, to our knowledge, no recent studies has probed into the BVOC emissions during the MH, despite about two decades ago Adams et al. (2001) proposed that such an exploration could significantly enhance our understanding of past climate change.

**3.2 Reduced dust emission**

In order to verify our model simulation results, especially concerning new dust emission configurations, the modelled dust deposition fluxes were compared with the reconstructed data (Fig. 3). For the PI period, the model results agree well with the proxy data at GC49 and 103GGC (Fig. 3a). At ODP658C, the model result is only 59% of the proxy data (Table 3). However, it should be noted that the dust flux at ODP658C may also include fluvial/shelf-derived flux, estimated to be about
34% of the actual dust flux as seen at the nearby site GC68 (see Fig. 3 in McGee et al., 2013). Consequently, the estimated actual dust flux at ODP658C during the PI is about 11.87 g m$^{-2}$ a$^{-1}$, closely aligning with our model result 10.63 g m$^{-2}$ a$^{-1}$. The



model result at VM20-234 (Fig. 3a) should not be viewed as an overestimation, but rather than consistent with the increasing trend of dust flux as we approach the PI era from 0.97 ka (see Fig. 2D in Williams et al., 2016).

During the MH period, the model results of both mh and mh_gsrd consistently reproduce the low dust fluxes at the remote sites, such as VM20-234, 103GGC, GGC3 and GGC6, despite underestimating the proxy data at VM20-234 (Fig. 3b). For the sites GC37, GC49 and GC68 along the northwest Africa coast, the model results in the mh case are consistent with the proxy data at GC37, but highly overestimates the values at GC49 and GC68 by about 3 to 4 times. The mh_gsrd case, on the other hand, matches with the proxy data at GC49 and GC68, but underestimates the value at GC37. This indicates that the
reduction of dust emissions due to the existence of water bodies over the region WSC may be overestimated in the mh_gsrd simulation. As for ODP658C, as mentioned above, the reconstructed dust flux may exceed the actual value, and here the actual flux is estimated as 1/2 to 1/3 of the original value based on the observation at sites GC66 (19.944° N, 17.860° W) and GC68 (see Fig. 3 in McGee et al., 2013), yielding an estimate of about 2.91 to 4.36 g m$^{-2}$ a$^{-1}$. Therefore, based on the estimation, while the mh case drastically overestimates the proxy data, the mh_gsrd case underestimates the value but agrees
better than mh. The comparison results affirm that the new dust emission configuration in the model is able to predict the dust flux pattern during both PI and MH periods, as demonstrated by the pi_ctrl and mh_gsrd cases respectively, across a substantial expanse stretching from the west Atlantic Ocean to the northwest coast of Africa.

Figure 4a shows that during the PI, the simulated dust emission occurs over the entire NA, especially the Sahara region,
reaching over 100 g m$^{-2}$ a$^{-1}$ in the regions such as LC, WSC and northern Algeria. However, during the MH the dust emission decreased significantly, especially in the paleolake regions LC and WSC, where the total annual dust emission is 2.9 (2.8) Tg a$^{-1}$ and 5.2 (25.2) Tg a$^{-1}$ in mh_gsrd (in mh), respectively (Table 2).

The values of mh align closely in magnitude with the model results from Egerer et al. (2018) for 6 ka, but the annual dust
emission in WSC region from the mh_gsrd case is about only one-tenth of that over similar Western Sahara region in Egerer et al. (2018). This descrepancy could result from that besides paleo lake configuration we also set the FPAR values with reconstructed vegetation cover which reduced the local dust emissions further. Later, when the paleo lakes eventually dried out, the exposed silt soil materials became vulnerable to surface wind, transforming the surrounding area into a major dust source (Tegen et al., 2002) This led to increased dust emission during the PI period, resulting in 80.1 Tg a$^{-1}$ over LC region
and 82.1 Tg a$^{-1}$ over WSC region under the pi_ctrl case (Table 2). Consequently, dust emissions are reduced by about 96% and 94% in the LC and WSC regions, respectively. Considering the dust emissions in the default PI case pi_orig are even higher than pi_ctrl (Fig. 4b), the relative reduction of dust emissions during the MH period compared to the PI period could be more pronounced.





*Figure 3: Reconstructed (circles) and modelled (square for the case pi_ctrl, triangle for the case mh and cross for the case mh_gsrd) dust deposition fluxes during (a) PI and (b) MH periods. Different colors are set for different sediment sites which are shown in the legend. The x axis shows the age in the unit of ka. The error bars of dust fluxes are plotted for GC37, GC49, GC68, VM20-234 and 103GGC, and the age error bars are plotted for GC37, GC49 and GC68.*




*Figure 4: Global annual dust emission in the unit of g m⁻² a⁻¹ over central and northern Africa in the cases pi_ctrl (a), mh (c) and mh_gsrd (e), as well as the dust emission differences between pi_orig and pi_ctrl (b), mh and pi_ctrl (d), mh_gsrd and pi_ctrl (f), respectively.*



The reduced dust emissions thus lead to a wide range of reduced surface dust concentration and dust load due to air-mass transport over NA during the MH period (Fig. 5). The most notable reduction in dust load in both the mh and mh_gsrd cases occurs over the LC and WSC regions, reflecting the dust emission differential pattern (Fig. 5d, f). The surface dust concentration closely correlated with dust load. When compared to pi_ctrl, the annual mean of these two variables over NA both declined by 63% in mh and 82% in mh_gsrd, respectively. And interestingly, the relative reduction of dust load in the mh_gsrd case corresponds with the prescribed reduction percentage of dust mixing ratio below 150 hPa over NA in Pausata et al. (2016). This report provided the original MH climate forcing conditions for the dynamic vegetation simulations in Lu et al. (2018).

## 3.3 Increased SOA

In TM5-MP, isoprene and monoterpenes are the sole precursors of SOA formation (Bergman et al., 2022). Therefore, the modelled surface concentration of SOA mirrors the emission rates of isoprene and monoterpenes in individual cases (Fig. 6a, c, e).

In the pi_ctrl case, the peak SOA surface concentration appears in central Africa along the longitude 20° E from around 0 to 10° N, with the highest values exceeding 2.5 μg m$^{-3}$ (Fig. 6a). In the mh case, SOA surface concentration increases due to higher BVOC emissions over the Sahel region with the highest recorded values exceeding 3.0 μg m$^{-3}$ (Fig. 6c). Moreover, the northernmost region of maximum concentration extends further in this case compared to pi_ctrl, but it is primarily confined to a narrow belt between 10° N and 20° N, reflecting the less northward extention of BVOC emissions. Here the average relative increase of SOA surface concentration over the NA region is 50% compared to pi_ctrl (Fig. 6d).

While in the mh_gsrd case, as the vegetation covers nearly the whole Africa, especially the NA region compared to the other two cases, the SOA surface concentration is higher than 4.5 μg m$^{-3}$ across western to central Africa (Fig. 6e). The significant increased BVOC emissions due to northward extension of vegetation cover result in more than 1 μg m$^{-3}$ SOA surface concentration nearly over the entire NA in mh_gsrd compared to pi_ctrl The difference rises to over 2 μg m$^{-3}$ across the western and central areas, resulting in as 2.9 times high as that in the pi_ctrl case over the NA region (Fig. 6f). Figure 6b shows that SOA surface concentration is lower in Central Africa in pi_ctrl compared to original PI simulation case pi_orig. This indicates that over this area, the simulated SOA concentration during the MH period is lower than that in the pi_orig case due to lower vegetation cover.





Figure 5: Same as Figure 2 but for dust load in the unit of mg m$^{-2}$.



*Figure 6: Same as Figure 2 but for SOA surface concentration in the unit of ug m$^{-3}$.*





### 3.4 Impact on size distributions and CCN

In order to estimate the potential impact of the increased SOA formation over NA region on cloud properties, we examined

the CCN surface concentrations at supersaturations of 0.2% (CCN0.2) (Fig. 7), which are roughly related to the number concentrations of particles larger than 100 to 200 nm (Jokinen et al., 2015). Within this size range, the CCN0.2 concentration is mainly determined by particles in the soluble accumulation (ACS) mode and solube coarse (COS) mode (Fig. S1 and S2). Nevertheless, during the PI period, anthropogenic primary aerosol emissions from Europe enhanced the particle number concentration in ACS mode and consequently the CCN0.2 concentration (Fig. 7a, b), exceeding the impact of land cover

change during the MH. Therefore, we conducted another simulation pi_zero which is the same as pi_ctrl but without any anthropogenic emissions. Compared to pi_zero, the average CCN0.2 surface concentration over NA in mh and mh_gsrd increased by about 7% and 17%, respectively.

The reduced dust emissions during the MH period decreased the mass concentration of COS mode over the NA region, but

the number concentration was unaffected (Figs. S1 and S2). Concurrently, over the same region, the number concentration of ACS mode increased (Figs. 7e and h) due to the enhanced SOA concentration which exceeded the impact of decreased sulfate concentration (Figs. S1 and S2), leading to increased CCN0.2 concentration (Figs. 7d and g). However, the hygroscopicity parameter of SOA is only 0.1 while for sulfate it is 0.6, so the CCN0.2 number concentration only increased less than 10 cm$^{-3}$ and 20 cm$^{-3}$ over the NA region in mh and mh_gsrd, respectively. It should be noted that the enhanced

BVOC emissions not only increased SOA concentration, but also consumed more OH (hydroxyl radical) via the oxidation reactions. Figures 7f and i show that the OH number concentration over the NA region in mh and mh_gsrd decreases more than 20% and 60% compared to the case pi_zero, respectively. This leads to reduced gas-phase sulfuric acid concentration which is produced mainly via the reaction between $SO_2$ (sulfur dioxide) and OH, and finally leads to lower sulfate concentration.


Figures 7e and h show a decrease of ACS mode in Central Africa in the mh and mh_gsrd cases, yet the CCN0.2 number concentration still shows positive change. The reduction in the number concentration of the ACS mode mainly attribute to the decreased BC (black carbon) and POM (primary organic matter) (Figs. S1 and S2), which have hygroscopicity parameters of 0 and 0.1, respectively. Therefore, although the number concentration of ACS mode decreases, the average

hygroscopicity of the particles could still increase due to the significant enhancement of SOA concentration over this region. Moreover, the average diameter of the ACS mode may not be affected due to the counteraction between the increased SOA concentration and the decreased concentrations of BC and POM.



**Figure 7:** *The difference of (a) CCN at supersaturation of 0.2% in the unit of # cm$^{-3}$, (b) number concentration of particles in the soluble accumulation mode (N_ACS) in the unit of # cm$^{-3}$ between pi_ctrl and pi_zero. (c) The relative change of OH surface number concentration in pi_ctrl compared to pi_zero. The same for (d) (g) CCN, (e) (h) N_ACS and (f) (i) OH relative change but between mh and pi_zero, mh_gsrd and pi_zero, respectively.*

In general, the enhanced BVOC emissions due to vegetation cover change are able to affect the cloud properties (e.g., reflectivity, lifetime and precipitation properties) via increasing CCN concentrations over the NA region, which needs to be kept in mind in future studies. Moreover, because of the complex consequences originated from the changed BVOC emissions as mentioned above, one should be careful when analyzing the aerosol-cloud interactions.





### 3.5 Reduction in AOD

During the PI period, the aerosol optical depth at 550 nm (AOD550) over NA is dominantly affected by the dust load (Fig.

8a and Fig. 5a). The dust emission source areas surrounding LC region show the highest aerosol optical depth, exceeding 0.3. During the MH, these areas along with their downwind locations show the largest reduction of AOD550 at around -0.3 in both mh and mh_gsrd cases (Fig. 8d, f). Both cases demonstrate similar patterns of AOD550 variation over NA. Interestingly, although the dust load in mh_gsrd is about 51.9% lower than that in mh over NA region, the AOD550 in mh_gsrd remains 0.01 higher which is about 8.2% increase compared to that in mh. This suggests that the AOD550 increase

due to the increasing of SOA surpasses the AOD550 reduction due to the decrease in dust. For example, in the mh case, the AOD550 over the NA region is split between 0.048 (51.6%) for dust and 0.045 (48.4%) for SOA. While in the mh_gsrd case, the values shift to 0.026 (25.2%) and 0.077 (74.8%), respectively. Given that mh_gsrd is the more realistic case, neglecting the AOD550 change due to the increased SOA over NA during the MH would result in underestimating the AOD550 by 0.045 (i.e., 0.077 minus 0.032). This accounts for about 21.2% of AOD550 in pi_ctrl case. Our result of

AOD550 for dust over the NA region in mh_gsrd is 0.026, which aligns with the simulation result of "MH control" (0.02 over Sahara) as presented in Thompson et al. (2019).

### 4 Conclusion

In this study, we produced a new dataset of emissions of isoprene and monoterpenes, and we also refined the methodology for calculating the dust emissions over Northern Africa during the mid-Holocene period. The updated emission data were

derived from the simulated vegetation cover data from Lu et al. (2018), which was found to be closely aligned with the reconstructed proxy data in Hély et al. (2014). Due to the northward expansion of grassland and trees, the total annual emission rate over the NA region increased from 27.0 Tg a$^{-1}$ during the PI period (pi_ctrl) to 114.8 Tg a$^{-1}$ during the MH period (mh_gsrd) for isoprene, and from 2.3 Tg a$^{-1}$ to 8.0 Tg a$^{-1}$ for monoterpenes. When compared with results from previous studies and PMIP4 simulations, our new dataset more accurately reflects the land cover change during the MH

which has been shown in Lu et al. (2018). Moreover, our finding of increased BVOC emissions during the MH period aligns with the results from Adams et al. (2001), who also calculated BVOC emissions from reconstructed vegetation cover.

Furthermore, the increased vegetation cover over Northern Africa resulted in a decreased annual dust emissions within this area from 280.6 Tg a$^{-1}$ during the PI period (pi_ctrl) to 26.8 Tg a$^{-1}$ during the MH period (mh_gsrd). Of the reduction, the LC

and WSC regions contributed to 77.2 Tg a$^{-1}$ and 76.9 Tg a$^{-1}$, respectively, altogether accounting for roughly 60.7% of the total reduction. It is noteworthy that these two regions (LC and WSC) were marked by the presence of actual water bodies, instead of preferential dust emission spots covered by silt and clay soil materials (Tegen et al., 2002; Egerer et al., 2018). Our model configuration of dust emissions was also evaluated by the agreement between the modelled and reconstructed dust deposition fluxes at different marine sediment locations across the Atlantic Ocean.





*Figure 8: Same as Figure 4 but for aerosol optical depth at 550 nm.*





The decreased dust emissions in mh_gsrd resulted in a significant 82.4% reduction in dust load and a comparable 82.3% reduction in dust AOD550. However, enhanced SOA formation from increased BVOC emissions also increased the AOD550, which ultimately caused a lesser total AOD550 reduction of 37.7%. Furthermore, the increased SOA surface

concentration and load increased the CCN0.2 number concentration near the surface over the NA region by 7% and 17% in mh and mh_gsrd cases, respectively, compared to the PI period This could potentially influence the regional climate via aerosol-cloud interactions. Nevertheless, our findings suggest that this may not be an efficient feedback pathway due to the counteracting effects originated from enhanced BVOC emissions.

Additionally, we have not analyzed the impact of the changing meteorology during the MH compared to present-day over NA, which is beyond the scope of this study. Nevertheless, it should be kept in mind that a reduced eastward wind pattern over NA as demonstrated by previous studies (e.g., Pausata et al., 2016; Williams et al., 2020) and increased rainfall could alter the transportation and wet deposition processes. Hence, more comprehensive investigations using coupled Earth system model simulations are needed to quantify how these processes altogether are impacting the precipitation and regional

climate.

Considering BVOC's critical role in the atmosphere-biosphere interactions as precursors of SOA (Kulmala et al., 2014; Yli-Juuti et al., 2021), more consistent BVOC emissions and consequently SOA formation will improve the simulation of the MH period, especially over Northern Africa in future model experiments. The findings of this study serve as a valuable

reference point for the emissions of isoprene, monoterpenes and dust in future MH model simulations, as will be defined in, e.g., PMIP5.

**Acknowledgement**

We acknowledge University of Helsinki Three Year Grant AGES, eSTICC (272041), the Academy of Finland Centre of Excellence (307331), the funding from EU H2020 project FORCeS (grant agreement No 821205), the ACCC Flagship

funded by the Academy of Finland (337549), the Autumn 2020 Arctic Avenue (spearhead research project between the University of Helsinki and Stockholm University), the European Commission Horizon Europe project FOCI (Non-CO2 Forcers and Their Climate, Weather, Air Quality and Health Impacts, grant agreement No 101056783), the project GRASS (Green Sahara: a cross-disciplinary approach to modelling climate and human distribution) funded by the Academy of Finland. We also thank CSC (IT Center for Science, Finland) for computational resources. Qiong Zhang acknowledges the

support from the Swedish Research Council (Vetenskapsrådet, Grant No 2017-04232). Additional funding was provided to Zhengyao Lu by the Swedish Research Council Formas (Grant No. 2020-02267), and the Swedish Research Council Vetenskapsrådet (Grant No. 2022-03617). The LPJ-GUESS data processing was enabled by resources provided by the National Academic Infrastructure for Supercomputing in Sweden (NAISS) and the Swedish National Infrastructure for



Computing (SNIC) partially funded by the Swedish Research Council through grant agreements no. 2022-06725 and no.
435 2018-05973.

**Author contributions**

RM and PTZ proposed the paper idea. PTZ processed the datasets, conducted the model configurations and simulations, analyzed the results, plotted the figures, and wrote the manuscript. ZYL provided the LPJ-GUESS data, and contributed to processing the data. JPK contributed to model configurations and simulations on CSC supercomputers. JL, TVN, PLS
contributed to model configurations. All authors provided comments, and contributed to paper discussion and writing.

**Competing interests**

The contact author has declared that none of the authors has any competing interests.

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
