# Peer review of "Simulating the dust emissions and SOA formation over Northern Africa during the mid-Holocene Green Sahara period"

_EGUsphere, 2023_

## Author Comment (AC1)

**Responses to Reviewer 1**

**Through model simulations, Zhou et al. found that although dust emission is greatly reduced during the mid-Holocene Green Sahara periods compared to PI, the emission of isoprene and monoterpenes increased substantially (~4 fold). The increased emission of these biogenic volatile organic compounds (BVOCs) enhanced the formation of secondary organic aerosol (SOA), eventually caused a 17% increase in cloud condensation nuclei. Such processes have been overlooked in almost all previous studies. The study is thus quite novel and the results have important implications to the climate impact of a green Sahara, and should be a useful contribution to the journal Climate of Past. The manuscript is well organized and written and the methods and results are reasonable, I suggest a minor revision.**

We really appreciate your patient review and helpful comments. We replied to each comment below and modified the manuscript accordingly.

**Comments:**

**1. In paragraph between lines 36-47, the influence of lakes on precipitation and west African monsoon should be mentioned. A few studies have worked on this, for example, Specht et al. (2022; https://doi.org/10.5194/cp-18-1035-2022) and Chandan and Peltier (2020; https://doi.org/10.1029/2020GL088728).**

Response: Thanks for pointing out this. We added the following text before "Other hypotheses ...":

"Besides, the larger lake and wetland extents over Northern Africa during the MH could also enhance the WAM and precipitation as demonstrated in previous studies (e.g., Krinner et al., 2012; Specht et al., 2022). When including the feedbacks from vegetation, soil and surface lakes in the model simulations for the MH, Chandan et al. (2020) demonstrated that the modelled precipitation enhancement was sufficient to maintain the extended vegetation cover over the Sahel and Sahara regions, even without changing dust forces."

**2. L94: "formextremely" misses a space.**

Response: The typo has been fixed:
"formextremely" -> "form extremely"

**3. The model was run for only two years, have the authors tested and checked that the results were stable already at the second year?**

Response: Since this question was raised by both reviewers, we replied in the same way as shown below:

We have conducted a 12-year simulation for the year 2009, in which the input data of 2009 were applied repeatedly for every simulation year. We found that for short-lived trace gases like monoterpenes and hydroxyl radical (OH), half a year was enough for the spin-up. For long-lived trace gases like methane (CH4), which are either prescribed or strongly constrained in the model, 7 to 8 months were enough for the spin-up (Fig. R1). We note that one-year spin-up for TM5-MP was also applied in previous studies (e.g., Williams et al., 2017; Myriokefalitakis et al., 2020). Therefore, we can assume that the results are already stable in the second year. And we added a sentence to clarify it and added Fig. R1 as Fig. S1.

"Each simulation ran for two years, with the first year serving as the spin-up and the results from the second year were analyzed."
->
"Each simulation ran for two years, with the first year serving as the spin-up and the results from the second year were analyzed. The one-year spin-up in TM5-MP simulations was also applied in previous studies (e.g., Williams et al., 2017; Myriokefalitakis et al., 2020), and was validated here in a 12-year simulation test case (see Fig. S1)."

Meanwhile the figure order in the supplement were also changed as:
Fig. S1: Time series of 12-year simulation for testing spin-up.
Fig. S2: Annual mean BVOC emissions during PI.
Fig. S3: Comparison of various aerosol quantities between mh and pi_zero.
Fig. S4: Comparison of various aerosol quantities between mh_gsrd and pi_zero.

**4. L162: "a-1" -> a$^{-1}$**

Response: The typo has been fixed:
"a-1" -> "a$^{-1}$"

**5. L191: I don't understand what "the latter one" refers to.**

Response: Here "the latter one" refers to the "eolian terrigenous flux". We have modified the sentence to make it clear:
"Therefore, we only compared the latter one with the model results (McGee et al., 2013)."
->
"Therefore, we only compared the eolian terrigenous flux with the modelled dust flux (McGee et al., 2013)."

**6. Figure 2, are the simulated emissions in PI reasonable compared to the observations? It would be useful if the authors can show the comparison between the pi_orig and the modern observations in the supplementary material.**

Response: As far as we know, there is no direct observation of BVOC emissions during PI currently, so we can only estimate them via model simulations. Here we added a comparison between our model results and a recent study conducted by Weber et al. (2022a). The BVOC emission data were obtained from Weber et al. (2022b). A new figure (Fig. R2, namely Fig. S2 in the supplement) with the analysis text was added in a new section in the supplementary as shown below:

"3. BVOC emissions during the pre-industrial (PI) period

In order to compare our modelled BVOC emissions during the PI period to other studies, we obtained the 30-year simulated emission data from a recent study by Weber et al. (2022a, b) (labelled as Weber2022 below), in which the iBVOC emissions system was applied to calculate the emissions of isoprene and monoterpenes during the PI (for more details refer to Weber et al., 2022a). Fig. S2 shows the comparison results over Africa between the cases pi_ctrl, pi_orig and Weber2022. The BVOC emissions data in pi_orig were taken from the MEGAN-MACC datasets for the year 2009, so it actually represents a present-day condition. The cases pi_orig and Weber2022 show similar emission patterns for both isoprene and monoterpenes emissions. The essential difference is that pi_orig shows larger latitudinal gradient than Weber2022. In pi_ctrl, the isoprene emission is more homogeneous in Central and Southern Africa compared to the other two cases, both of which show an apparent higher emission over Central Africa. For monoterpenes,

pi_ctrl shows higher emissions in Southern Africa and lower emissions in Central Africa, while the other two cases present an opposite pattern. We should note that over Northern Africa, which is the domain of focus of this study, all of these PI cases show low to none emissions of isoprene and monoterpenes. It indicates that our main results and conclusions are not affected by the uncertainties of PI BVOC simulations."

**7. L270-282: Table 3 should cited in this paragraph**

Response: We added the citation to Table 3 and Fig. 3b in several places as shown below:

"at GC49 and GC68 by about 3 to 4 times."
->
"at GC49 and GC68 by about 3 to 4 times (Table 3)."

"but underestimates the value at GC37."
->
"but underestimates the value at GC37 (Fig. 3b)."

"the mh_gsrd case underestimates the value but agrees better than mh."
->
"the mh_gsrd case underestimates the value but agrees better than mh (Table 3)."

**8. L323: "significant" -> "significantly"**

Response: The typo has been fixed:
"significant" -> "significantly"

**9. In the captions of Figures 5&6, do the authors mean "same as Figure 4"?**

Response: Thanks for pointing out it. Yes, they should be "same as Figure 4". The caption texts have been modified:
"Figure 5: Same as Figure 2" -> "Figure 5: Same as Figure 4"
"Figure 6: Same as Figure 2" -> "Figure 6: Same as Figure 4"

**10. L125: It will be better if the authors directly state here that the present-day meteorological data were used in all 5 experiments. Although the sentence implies this but I assume it was only for the three PI experiments, and did not realize this until the end of the manuscript. This is a caveat of the study because impact of BVOC on CCN might be different if the MH meteorological data (some of the authors clearly have such data from model simulations) had been used.**

Response: We have modified the text to make the statement more explicit:
"Unless otherwise specified, the input meteorological data ..."
->
"In all the five simulation cases, the input meteorological data ..."

And we agree with the reviewer that the changing meteorological input in MH would influence the impact of BVOC on CCN and also the SOA formation. However, firstly, from the technical point of view, the meteorological data of MH that we obtained from other model simulations were not compatible with the TM5-MP input, so they can not be applied in a straightforward way. Secondly, in this study we plan to first quantify the one-way impact of land surface change on SOA formation and CCN concentration. The more complete feedbacks between land surface change and climate

can be simulated in the Earth system model, which is beyond the scope of this study. Therefore, we added a paragraph to point it out at L410-415 in the manuscript.

[Figure]

*Figure R1: Time series of (a) tropospheric mean mixing ratio of methane (CH4), (b) tropospheric mean mixing ratio of monoterpenes and (c) tropospheric mean number concentration of hydroxyl radical (OH) in a 12-year simulation. The input data of 2009 were applied repeatedly for each year, but the tick labels show increasing year numbers to represent the continuous simulation years. Here the tropospheric mean values are calculated over the first 21 model layers, in which the global mean air pressure of the top layer (layer 21) is around 200 hPa.*

[Figure]

*Figure R2: Annual mean emission rates of isoprene (emiisop) over Africa from the simulation cases (a) pi_ctrl, (b) pi_orig and (c) the emission data from Weber et al. (2022a, b) (Weber2022). The annual mean emission rates of monoterpenes (emiterp) are also shown for simulation cases (d) pi_ctrl, (e) pi_orig and (f) Weber2022.*

**References:**

Chandan, D., and Peltier, W. R.: African Humid Period Precipitation Sustained by Robust Vegetation, Soil, and Lake Feedbacks. Geophysical Research Letters, 47(21). https://doi.org/10.1029/2020GL088728, 2020.

Krinner, G., Lézine, A.-M., Braconnot, P., Sepulchre, P., Ramstein, G., Grenier, C., and Gouttevin, I.: A reassessment of lake and wetland feedbacks on the North African Holocene climate. Geophysical Research Letters, 39(7). https://doi.org/10.1029/2012GL050992, 2012.

Myriokefalitakis, S., Daskalakis, N., Gkouvousis, A., Hilboll, A., Van Noije, T., Williams, J. E., Le Sager, P., Huijnen, V., Houweling, S., Bergman, T., Rasmus Nüß, J., Vrekoussis, M., Kanakidou, M., and Krol, M. C.: Description and evaluation of a detailed gas-phase chemistry scheme in the TM5-MP global chemistry transport model (r112). Geoscientific Model Development, 13(11), 5507–5548. https://doi.org/10.5194/gmd-13-5507-2020, 2020.

Specht, N. F., Claussen, M., and Kleinen, T.: Simulated range of mid-Holocene precipitation changes from extended lakes and wetlands over North Africa. Climate of the Past, 18(5), 1035–1046. https://doi.org/10.5194/cp-18-1035-2022, 2022.

Weber, J., Archer-Nicholls, S., Abraham, N. L. Shin, Y. M., Griffiths, P., Grosvenor, D. P., Scott, C. E., and Archibald, A. T.: Chemistry-driven changes strongly influence climate forcing from vegetation emissions. Nat. Commun., 13, 7202, 2022a.

Weber, J., Archer-Nicholls, S., Abraham, N. L., Shin, Y. M., Griffiths, P., Grosvenor, D. P., Scott,

C. E., and Archibald, A. T.: Research data supporting "Chemistry-driven oxidant changes strongly influence climate forcing from vegetation emission". Apollo - University of Cambridge Repository. https://doi.org/10.17863/CAM.83526. 2022b.

Williams, J. E., Folkert Boersma, K., Le Sager, P., and Verstraeten, W. W.: The high-resolution version of TM5-MP for optimized satellite retrievals: Description and validation. Geoscientific Model Development, 10(2), 721–750, 2017.

---

## Author Comment (AC2)

**Responses to Reviewer 2**

**Comments to the authors**

We really appreciate your patient review and helpful comments. We replied each comment below and modified the manuscript accordingly.

**Please delete the following texts in the section 2.2, as I do not think it is really relevant to the scientific issues discussed, but rather to the technical details of the model setup.**

**"The model was installed on Puhti supercomputer at CSC (IT Center for Science, Finland), and 90 CPU (Central Processing Unit) cores were utilized for each parallel simulation run. One simulation year cost about 10 hours in real life."**

Response: Thanks for point out this. We removed this sentence in the modified manuscript.

**Here The authors conduct two years run of TM5-MP:one year is used for spin up and the other is used for analysis. I am not sure if it is too short for one year simulation to calculate the mean state of variables.**

Response: Since this question was raised by both reviewers, we replied in the same way as shown below:

We have conducted a 12-year simulation for the year 2009, in which the input data of 2009 were applied repeatedly for every simulation year. We found that for short-lived trace gases like monoterpenes and hydroxyl radical (OH), half a year was enough for the spin-up. For long-lived trace gases like methane ($CH_4$), which are either prescribed or strongly constrained in the model, 7 to 8 months were enough for the spin-up (Fig. R1). We note that one-year spin-up for TM5-MP was also applied in previous studies (e.g., Williams et al., 2017; Myriokefalitakis et al., 2020). Therefore, we can assume that the results are already stable in the second year. And we added a sentence to clarify it and added Fig. R1 as Fig. S1.

"Each simulation ran for two years, with the first year serving as the spin-up and the results from the second year were analyzed."
->
"Each simulation ran for two years, with the first year serving as the spin-up and the results from the second year were analyzed. The one-year spin-up in TM5-MP simulations was also applied in previous studies (e.g., Williams et al., 2017; Myriokefalitakis et al., 2020), and was validated here in a 12-year simulation test case (see Fig. S1)."

Meanwhile the figure order in the supplement were also changed as:
Fig. S1: Time series of 12-year simulation for testing spin-up.
Fig. S2: Annual mean BVOC emissions during PI.
Fig. S3: Comparison of various aerosol quantities between mh and pi_zero.
Fig. S4: Comparison of various aerosol quantities between mh_gsrd and pi_zero.

[Figure]

*Figure R1: Time series of (a) tropospheric mean mixing ratio of methane (CH4), (b) tropospheric mean mixing ratio of monoterpenes and (c) tropospheric mean number concentration of hydroxyl radical (OH) in a 12-year simulation. The input data of 2009 were applied repeatedly for each year, but the tick labels show increasing year numbers to represent the continuous simulation years. Here the tropospheric mean values are calculated over the first 21 model layers, in which the global mean air pressure of the top layer (layer 21) is around 200 hPa.*

**References:**

Myriokefalitakis, S., Daskalakis, N., Gkouvousis, A., Hilboll, A., Van Noije, T., Williams, J. E., Le Sager, P., Huijnen, V., Houweling, S., Bergman, T., Rasmus Nüß, J., Vrekoussis, M., Kanakidou, M., and Krol, M. C.: Description and evaluation of a detailed gas-phase chemistry scheme in the TM5-MP global chemistry transport model (r112). Geoscientific Model Development, 13(11), 5507–5548. https://doi.org/10.5194/gmd-13-5507-2020, 2020.

Williams, J. E., Folkert Boersma, K., Le Sager, P., and Verstraeten, W. W.: The high-resolution version of TM5-MP for optimized satellite retrievals: Description and validation. Geoscientific Model Development, 10(2), 721–750, 2017.